# High precision PINNs in unbounded domains: application to singularity formulation in PDEs

## Abstract

We investigate the high-precision training of Physics-Informed Neural Networks (PINNs) in unbounded domains, with a special focus on applications to singularity formulation in PDEs. We propose a modularized approach and study the choices of neural network ansatz, sampling strategy, and optimization algorithm. When combined with rigorous computer-assisted proofs and PDE analysis, the numerical solutions identified by PINNs, provided they are of high precision, can serve as a powerful tool for studying singularities in PDEs. For 1D Burgers equation, our framework can lead to a solution with very high precision, and for the 2D Boussinesq equation, which is directly related to the singularity formulation in 3D Euler and Navier-Stokes equations, we obtain a solution whose reported training loss is about four digits below the previously reported PINN computation under our training protocol. Because the exact implementation and collocation points of the previous work are not publicly available, we treat this comparison as a quantitative reference rather than a fully controlled side-by-side benchmark. We also discuss potential directions for pushing towards machine precision for higher-dimensional problems.

## 1 Introduction

Singularity formulation is one of the key challenges in the study of partial differential equations (PDEs). Unlike well-posed equations, where one can apply classical existence and uniqueness theorems, singularities often occur in certain solutions of nonlinear PDEs, where we only have guarantees of existence for a short time, but the solution may blow up in finite time. The study of singularities often involves a case-by-case approach and is related to some of the most intriguing mathematics and physical properties, such as the onset of turbulence in the Navier-Stokes equations. The singularity of the Navier-Stokes equations is one of the seven Millennium Prize Problems (Fefferman, 2006): widely regarded as the most fundamental and challenging problem in analysis, and is still open. One of the key difficulties in the study of singularities is the lack of understanding of the singularity pattern and its mechanism. The computation of the singularity itself, or infinity, is intractable numerically. A general roadmap is thus to first propose a plausible singularity ansatz that renders the computation feasible, then find candidates of such blowup by numerical simulations, and finally verify the stability of such an ansatz by PDE analysis.

We are often interested in a special structure of singularity: self-similar singularity. Self-similarity relates to the invariance of the solution under scaling transformations and reduces singularity to the existence of a self-similar profile. To be precise, for the quantity of interest $u(x,t)$, we put the ansatz $u(x,t) = (T-t)^{-\alpha}U(x(T-t)^{-\beta})$, where $U$ is the profile function independent of time, $T$ is the blowup time, and $\alpha > 0, \beta$ are the scaling exponents to be determined. Now we can reduce the computation of an infinite $u$ to the computation of a finite, smooth profile $U$, along with scaling exponents $\alpha, \beta$ to be inferred. This will translate later to a profile equation with unknown scaling $\lambda$ proportional to $\beta/\alpha$; see (1) and for example (5) in the Burger's case. Physics-Informed Neural Networks (PINNs) (Raissi et al., 2019) serve as a powerful tool to find such profiles, with the scaling parameters jointly inferred as inverse problems. It was first introduced in the context of identifying singularity profiles in Wang et al. (2023) and has seen success even in problems that are unstable for time-marching in traditional numerical methods, indicating that the blowup profile that has unstable directions in the rescaled dynamics and therefore typically requires fine-tuning or codimension conditions,

in contrast to a stable attractor-like blowup regime. For these self-similar inverse-profile problems on very large effective domains, with symmetry, nondegeneracy, and far-field structure all needing to be enforced simultaneously, PINNs provide a flexible way to encode these constraints and jointly optimize the profile and scaling parameters. While PINNs offer a powerful tool to search for candidate profiles, solutions identified by PINNs are often of limited accuracy and far from applicable to rigorous PDE analysis, to the best of the authors' knowledge.

In this work, we aim to systematically study the high-precision training of PINNs, with a special focus on applications to solve profile equations governing singularity formulations in PDEs, especially in fluid dynamics. We will take a modularized perspective while avoiding highly problem-specific engineering choices and investigate the following aspects: a good neural network ansatz representing the profile function, a good sampling strategy to tackle the infinite domain with a special focus on imposing boundary conditions, and a good optimization algorithm to train the neural network. We apply our findings to the 1D Burgers equation and the 2D Boussinesq equation, obtaining a precision amenable to rigorous PDE analysis for the 1D Burgers equation and a Boussinesq profile whose reported training loss is about four digits below the previously reported PINN computation of Wang et al. (2023) under our training protocol. While Wang et al. (2023) established the overall PINN-based self-similar framework for these PDEs, our focus here is high-precision training on unbounded domains, especially the role of exact versus weak asymptotics, hard nondegeneracy constraints via Taylor's expansion, and self-scaled Broyden updates in pushing the residual significantly lower. We remark that a later work (Wang et al., 2025b) achieves even higher precision while discovering unstable blowups. The 2D Boussinesq equation shares many similarities with the 3D axisymmetric Euler equation for the ideal fluid without viscosity; see the pioneering works of (Chen et al., 2021; Chen & Hou, 2022; 2025) for the connection between the two equations, where the authors used the connection to establish a singularity formulation for the 3D axisymmetric Euler with boundary. We remark that although we numerically only evaluate the training loss, which is a population approximation of the equation residual after resampling, this could lead to a full proof and smallness of solution error, potentially due to the following two observations. Firstly, as observed in Wang & Lai (2024), with resampling, small population loss in PINNs indicates empirical generalization across the solution domain. Secondly, coupled with the stability analysis, one can upgrade the a posteriori smallness of equation residual to a priori estimates on the solution error; see again the Euler singularity (Chen et al., 2021; Chen & Hou, 2022; 2025) along with the recent computer-assisted proof of nonuniqueness of Leray-Hopf weak solutions to the Navier-Stokes equations (Hou et al., 2025).

## 2 Related Works

### 2.1 PINNs

Neural networks have witnessed success in solving PDEs and surrogate modeling in math and science. PINNs in particular have been widely used due to their flexibility and applicability to a wide range of problems (Cai et al., 2021; Raissi et al., 2019; Karniadakis et al., 2021). The key idea of PINNs is to enforce the PDE constraints at a set of collocation points and to minimize the residual of the PDEs as a loss function. By posing the solution of PDEs as an optimization problem, PINNs are especially suited to solve inverse problems (Raissi et al., 2020; Yuan et al., 2022; Lu et al., 2021b; Yu et al., 2022), where the solution of the PDE and the underlying parameters can be jointly inferred. In Wang et al. (2023), the authors used PINNs to study the blowup of the 1D Burgers equation, the 1D family of generalized Constantin-Lax Majda equations, and the 2D Boussinesq equation.

Another line of work, operator learning (Kovachki et al., 2023), focuses on learning the solution operator instead of learning a single instance of solution, where Fourier Neural Operators (FNOs) (Li et al., 2020; 2023a;b; Li et al.) and DeepONets (Lu et al., 2021a) are two families of representative works in this direction. Once a solution operator is learned, it can be evaluated in a resolution-free manner at any point in the domain. Data are often augmented to enhance the solution accuracy, while the loss function can also incorporate the PDE constraints, termed Physics Informed Neural Operator (PINO) (Li et al., 2021). In Maust et al. (2022), the authors used PINO with Fourier continuation to study the blowup of the 1D Burgers equation.

## 2.2 Self-similar Singularity and Computer Assisted Proofs

Self-similar singularity of the ansatz $u(x,t) = (T-t)^{-\alpha}U(x(T-t)^{-\beta})$ is generic in the study of singularity formation in PDEs, where one uses the scaling invariance of the PDE and can reduce the computation of an infinite $u$ to the computation of a finite, smooth (approximate) profile $U$. Such structures exist even in the simple Riccati ODE $u_t = u^2$ with an exact solution $u = (T-t)^{-1}$.

The approximate profile can be identified via explicit construction or numerical computation. Working in the rescaled, self-similar variables and performing stability analysis around the profile $U$ provides a powerful tool to establish the singularity formation, for nonlinear Schrodinger equations (McLaughlin et al., 1986; Merle & Raphael, 2005), incompressible fluids (Elgindi & Jeong, 2019; Chen & Hou, 2021; Chen et al., 2021; Chen & Hou, 2022; Hou & Wang, 2024), and compressible fluids (Merle et al., 2022a;b). Until recently, most of the works relied on an explicit profile and spectral information of the associated linearized operator to establish linear and nonlinear stability. In Chen & Hou (2022; 2025), the authors used computer-assisted proofs with a sophisticated numerical profile obtained by solving the dynamic rescaling equations in time to obtain an approximate steady state. By analyzing the stability of the approximate profile, they established the singularity formation for the 2D Boussinesq equation and the 3D axisymmetric Euler equation with boundary. In Hou et al. (2024); Chen et al. (2024a); Liu et al. (2025), the authors provided a framework using only local information for stability analysis, bypassing spectral information and allowing for numerical profiles with computer-assisted proofs, for problems beyond self-similarity.

## 2.3 Towards High Precision Training

Various methods have been proposed in the literature to improve the accuracy of PINNs. One line of work focuses on a better representation of the solution. In Michaud et al. (2023); Wang & Lai (2024), the boosting technique was proposed, where a sum of a sequence of neural networks with decreasing magnitude was used to learn the solution; at each stage, a new neural network is trained to learn the residual. To overcome the spectral bias (Rahaman et al., 2019) of multilayer perceptrons (MLPs), or the favor of learning low-frequency modes (Xu et al., 2019b;a) in the solution, one can use Fourier feature encoding (Sitzmann et al., 2020; Tancik et al., 2020; Ng et al., 2024), or different activation functions (Jagtap et al., 2020b;a; Hong et al., 2022; Zhang et al., 2023; Wang et al., 2024a). In particular, Kolmogorov-Arnold Networks (KANs) (Liu et al., 2024b;a) that leverage nonlinear learnable activation functions and the Kolmogorov-Arnold representation theorem were proposed and further investigated in the PINN setting (Wang et al., 2024b; Shukla et al., 2024; Toscano et al., 2025). Another line of work improves the optimization landscape during the training of PINNs. Various optimizers, which we will detail in Subsection 3.3, have been proposed to improve the convergence rate. Adaptive design of points sampling (Anagnostopoulos et al., 2024; Wu et al., 2023; Rigas et al., 2024) and adaptive weighting of different terms (Wang et al., 2021; Xiang et al., 2022; McClenny & Braga-Neto, 2023) in the loss function were also proposed to improve the accuracy of PINNs.

We will only focus on applying hard constraints and choosing a good optimizer in this work, and leave the exploration of more sophisticated tricks like boosting or adaptive sampling of collocation points for future work.

# 3 Methodology

We outline our methodology of high-precision training for PINNs on the whole space in this section. We work under the general formulation of the profile equation

$$L(U, \lambda) = 0, \tag{1}$$

where $U(y)$ is the profile function, $\lambda$ is a set of scaling parameters to be determined, and $L$ is the nonlinear differential operator. For our problems of interest, $U=0$ will be a trivial solution satisfying the equation.

### 3.1 Infinite Domain

The key challenges we are facing here are sampling and learning on an infinite domain. For a given budget of a finite number of sampling points, we need to sample the domain in a way that the resulting solution is accurate and generalizes well throughout the domain. In the meantime, we want the neural network to be able to represent the profile function and the initialization of parameters to favor learning of such representations. To this end, we adopt an exponential "mesh" in our sampling strategy: Consider an auxiliary variable $z$ such that $y = \sinh(z) = \frac{e^z - e^{-z}}{2}$, and sample $z$ uniformly in a finite region. Here we choose the sinh transformation as in Wang et al. (2023) to respect the parity of the functions, detailed in the subsequent subsection. Such a transformation maps roughly $z \in [-30, 30]$ to $y \in [-5 \times 10^{12}, 5 \times 10^{12}]$.

Boundary conditions are another important aspect when learning on the whole space. For our problems of interest, $U$ by itself will not have sufficient decay at infinity, and one approach adopted in Wang et al. (2023) is to impose Neumann boundary conditions at infinity, or numerically on the boundary of the domain of the $z$ variables. To rule out the trivial solution $U = 0$, we need to enforce a nondegeneracy condition, often posed at the origin. We will discuss the enforcement of these conditions in the following subsection. We refer to this formulation as boundary conditions using **weak asymptotics**.

On the other hand, we can enforce stronger information on the boundary. If we know the exact asymptotic behavior of the solution at infinity as $g$, for example a power law, we can introduce a smooth cutoff function $\chi$ with $\chi(0) = 0, \chi(\infty) = 1$ and the ansatz $U = \tilde{U} + \chi g$ for the asymptotics $g$. We can then enforce Dirichlet boundary conditions at infinity for $\tilde{U}$ represented by the neural network. We refer to this formulation as boundary conditions using **exact asymptotics**. We will demonstrate for the 1D example that PINNs using exact asymptotics will outperform those using weak asymptotics by a large margin.

A priori, the exact asymptotics information is not available, and one can first train a neural network $U_w$ with boundary conditions using weak information, and distill the information of asymptotics $g$ from $U_w$. We refer to this formulation as boundary conditions using **hybrid asymptotics**. For example, for the 2D Boussinesq equation, borrowing ideas from Chen & Hou (2022), one can use function fitting and symbolic regression to extract asymptotics $g$ from $U_w$, filtering out the noisy residual, such that $g$ is a symbolic function approximating $U_w$ at infinity. We will leave this approach to future work.

### 3.2 Hard Constraint

Hard constraints are important concepts in the parametrization of the solution space for PINNs. When enforced properly, they will guarantee physical properties of the solution (Lu et al., 2021b; Richter-Powell et al., 2022; Mohan et al., 2023; Duruisseaux et al., 2024), and can impose the solution to be in the correct manifold. Previous works focus on hard constraints on the boundary. For unbounded domains, analogous conditions arise as nondegeneracy and parity.

**Parity.** For a function $f(y_i, \hat{y}_i)$ even/odd in the variable $y_i$, we train a neural network with the following ansatz $f = (f_{nn}(y_i, \hat{y}_i) \pm f_{nn}(-y_i, \hat{y}_i))/2$. Parities exist here due to the physics of the singularities.

**Nondegeneracy conditions.** As discussed in the previous subsection, we need to enforce nondegeneracy conditions to rule out the trivial solution $U = 0$ when using weak asymptotics. For example, for the 1D Burgers equation, we know that $U$ is odd and necessarily $U'(0) = -1$; we can enforce $U'''(0) = 6$ via a rescaling symmetry in space. We will enforce a hard constraint via Taylor expansion at the origin as $U = -z + z^3 + z^4 U_1$, for an odd function $U_1$. Similarly for the 2D Boussinesq equation, we enforce $\partial_1 \Omega(0,0) = -1$ and $\Omega$ is odd in $z_1$ via a Taylor expansion as $\Omega = -z_1 + z_1 z_2 \Omega_1 + z_1^2 \Omega_2$, where $\Omega_1, \Omega_2$ are even and odd functions in $z_1$ respectively. We remark that such hard constraints are essential to identify nontrivial solutions numerically and empirically soft constraints tend to fail. One can proceed similarly for other equations.

Empirically we observe a better convergence rate and a more stable solution when enforcing hard constraints.

### 3.3 Optimizer: Self-Scaled BFGS Methods

A common practice of training PINNs is to use the Adam optimizer. As a stochastic first-order method, Adam is known to be robust and efficient in training deep neural networks and can empirically escape local minima. To further improve convergence to the minimizer, one can apply second-order methods with a higher convergence rate, like L-BFGS, after training with Adam for a few epochs. While this seems to be a gold standard in the training of PINNs (Rathore et al., 2024), various optimizers have been investigated, including variants of second-order quasi-Newton methods (Rathore et al., 2024; Wang et al., 2025a), and optimizers using natural gradients (Müller & Zeinhofer, 2023; Jnini et al., 2024; Chen et al., 2024b). We highlight and use the self-scaled BFGS methods proposed in Al-Baali (1998); Al-Baali et al. (2014) and introduced to the PINNs context in Urbán et al. (2025); Kiyani et al. (2025). BFGS methods use an approximation of the inverse of the Hessian matrix $H_k$ to precondition the gradient for the update direction. To be precise, consider the parameters $\Theta_k$ and learning rate $\alpha_k$ at step $k$, with loss function $\mathcal{J}(\Theta)$, then the update rule for $\Theta$ is

$$\Theta_{k+1} = \Theta_k - \alpha_k H_k \nabla \mathcal{J}(\Theta_k).$$

Different choices of updating the approximate inverse Hessian $H_k$ lead to different optimizers, and L-BFGS in particular is a memory-efficient way for the updates by storing only vectors instead of the whole matrix. The self-scaled BFGS methods use a scaling compared to the standard BFGS update of the inverse Hessian. More precisely, for the auxiliary variables

$$s_k = \Theta_{k+1} - \Theta_k, \quad y_k = \nabla \mathcal{J}(\Theta_{k+1}) - \nabla \mathcal{J}(\Theta_k),$$
$$v_k = \sqrt{y_k \cdot H_k y_k} \left[ \frac{s_k}{y_k \cdot s_k} - \frac{H_k y_k}{y_k \cdot H_k y_k} \right],$$

we have for the scaling factors $\tau_k$ and $\phi_k$:

$$H_{k+1} = \frac{1}{\tau_k} \left[ H_k - \frac{H_k y_k \otimes H_k y_k}{y_k \cdot H_k y_k} + \phi_k v_k \otimes v_k \right] + \frac{s_k \otimes s_k}{y_k \cdot s_k},$$

where the original BFGS corresponds to the choices $\tau_k = \phi_k = 1$. While this is only a simple modification of the original BFGS, the authors in Urbán et al. (2025) demonstrated a much improved convergence rate across a variety of benchmarks, including the Helmholtz equation, the nonlinear Poisson equation, the nonlinear Schrödinger equation, the Korteweg-De Vries equation, the viscous Burgers equation, the Allen-Cahn equation, 3D Navier-Stokes: Beltrami flow, and the lid-driven cavity. We use the self-scaled Broyden methods proposed in Urbán et al. (2025); see equations (13)-(23) therein for details on the choices of $\tau_k$ and $\phi_k$. We remark that we observe using the traditional Adam or LBFGS optimizers empirically would result in losses plateauing at much higher values.

**On the role of minibatch training or random resampling.** One of the common practices when training PINNs is to use random resampling of the collocation points. This can enhance the performance of SGD-based methods like Adam empirically. However, full-batch second-order methods like BFGS with supposedly higher-order accuracy do not adapt well to random resampling since they rely on past trajectories for Hessian updates. One empirical observation, as proposed in Wang & Lai (2024), is that when one uses an optimizer with fixed resolution like BFGS, it will be able to generalize in the regions where sampling points are sufficient. However, in the undersampled regions, the learned solution generalizes poorly. In an abstract form, there exists a critical batchsize $N_c$, such that when $N > N_c$, fixed sampling will be preferred, while for $N < N_c$, fixed sampling will have very bad generalization. $N_c$ would depend on both the equation and the scale of the neural network. See details on the choices of the batch size in the experiments section. Empirically, we observe that roughly 10k points are sufficient for generalization with fixed training points in the Burgers experiments. We use periodic resampling every 1000 iterations as a practical compromise: the collocation set remains fixed long enough for the quasi-Newton history to remain meaningful, while occasional refreshing reduces dependence on one finite training grid.

## 4 Experiments

In this section, we describe our numerical experiments on the blowup profiles for 1D Burgers equation and 2D Boussinesq Equation. When training both equations, we denote the PDE by $L(U(y)) = 0$ and the boundary condition by $B(U) = 0$. We use auxiliary variables $z = \sinh^{-1} y$ as in Subsection 3.1 and consider the following combination of interior, boundary, and smoothness losses as in Wang et al. (2023)

$$
\begin{aligned}
\mathcal{J}(\Theta) &= 0.1(L_i + L_s) + L_b \\
&= 0.1(\frac{1}{N_i}\sum_{j=1}^{N_i}[\hat{L}(U_{nn}(z_j))]^2 + \frac{1}{N_s}\sum_{j=1}^{N_s}|\nabla_{z_j}\hat{L}(U_{nn}(z_j))|^2) + \frac{1}{N_b}\sum_{j=1}^{N_b}[\hat{B}(U_{nn})]^2,
\end{aligned}
\tag{2}
$$

where $U_{nn}(z)$ is supposed to approximate $\hat{U}(z) = U(y)$ in the $z$-variables, and $\hat{L}$, $\hat{B}$ denotes the PDE and the boundary condition transformed in the $z$-variables; see Wang et al. (2023) for a concrete formula for the 2D Boussinesq equation. We choose the coefficient 0.1 in Eq. (2) to balance the equation/smoothness losses and the boundary loss; this value is kept fixed across the reported experiments.

The experiments below are organized to separate the evidence for the main components as much as possible. The Burgers experiment gives the cleanest controlled ablation of the far-field treatment, since the architecture, optimizer, parity constraint, sampling distribution, and training budget are fixed. For the other components, especially hard nondegeneracy constraints, periodic resampling, and the Adam-to-SSBroyden optimization pipeline, the evidence is diagnostic rather than a complete factorial ablation. We remark that the experiments are reported on single runs.

Table 1: **Component-level evidence from the reported experiments.** The table makes explicit which comparison or diagnostic in the current experiments supports each modular component of the method.

| Component | Where it is isolated | Evidence reported |
| --- | --- | --- |
| Far-field asymptotics | Burgers weak vs. exact asymptotics | Figures 1–2 compare residuals and losses with all other settings fixed. |
| Hard constraints | Burgers and Boussinesq parametrizations | Parity and Taylor constraints enforce the nontrivial symmetry class and prevent convergence to the trivial solution. |
| Optimizer | Boussinesq Adam followed by SS-Broyden1 | Figure 5 shows the loss decrease after switching to the self-scaled Broyden stage. |
| Resampling | Fixed large collocation sets with periodic resampling | Resampling every 1000 iterations reduces overfitting while preserving quasi-Newton stability. |

### 4.1 Burgers Equation

For the 1D Burgers equation

$$
u_t + uu_x = 0,
\tag{3}
$$

consider the self-similar ansatz that respects the scaling symmetry

$$
u(x,t) = (1-t)^\lambda U(y), \quad y = x(1-t)^{-1-\lambda}.
\tag{4}
$$

The profile equation for $U$ used for the PDE loss in (2) is

$$
-\lambda U + ((1+\lambda)y + U)U_y = 0.
\tag{5}
$$

We impose an odd symmetry on $U$, and the profile equation has implicit solutions

$$
y + U + CU^{1+1/\lambda} = 0.
\tag{6}
$$

for any constant $C$, as in the setting of Wang et al. (2023), where we know that the most stable solutions correspond to $\lambda = 0.5$ and there are nonsmooth solutions at e.g. $\lambda = 0.4$.

In this example, we assume that we first train the neural network on a bounded domain and infer the correct $\lambda$ already, for example via the method in Wang et al. (2023). Now we focus on fixing $\lambda$ and learn $U$ on the unbounded domain. Using an MLP with activation function tanh, 4 layers and 20 neurons per layer and a hard constraint on parity, we use the optimizer SSBroyden1 as in Urbán et al. (2025) with 20000 epochs and resampling every 1000 epochs. $z$ is sampled uniformly on $[0, 30]$ with a batchsize 10000 for both the interior and smoothness losses, corresponding to a domain $[0, 5 \times 10^{12}]$ in the $y$ variables.

For the formulation using weak asymptotics as in Subsection 3.1, we use the Neumann boundary condition $U_y = 0$ and enforce hard constraint of nondegeneracy conditions as in Subsection 3.2. For the formulation using exact asymptotics as in Subsection 3.1, we use Dirichlet boundary condition $\tilde{U} = 0$ and the cutoff function $\chi = (\frac{y}{1+y})^{15}$, since the far field is captured by the exact asymptotics $g = -y^{\frac{\lambda}{1+\lambda}}$.

To address the question of whether a small residual corresponds to an accurate profile, we also compare the learned Burgers profile directly with the implicit reference solution. Writing $V = -U \geq 0$, the implicit relation can be evaluated as $y = V + V^{1+1/\lambda}$, which we solve pointwise on a held-out grid. Table 2 reports a reduced-budget diagnostic for the exact-asymptotics case with $\lambda = 0.5$, using one-tenth of the original batch size and one-tenth of the original evaluation grid. It reaches a small direct profile error and a small held-out residual. The residual $R$ is defined via the equation loss as $R(y) = -\lambda U(y) + ((1 + \lambda)y + U(y))U_y(y)$

Table 2: **Direct Burgers profile error.** We compare the exact-asymptotics Burgers PINN for $\lambda = 0.5$ with the implicit reference solution on a held-out grid. This diagnostic uses a reduced budget: 800 SSBroyden1 iterations, batch size 1000, and 2000 evaluation points on $z \in [0, 30]$. The periodic run resamples every 100 iterations, corresponding to one-tenth of the full-run resampling period.

| $\|U_{\mathrm{NN}} - U_{\mathrm{ref}}\|_\infty$ | Relative $L^2$ error | $\|R\|_\infty$ | RMS residual | Final loss |
|---|---|---|---|---|
| $1.895 \times 10^{-5}$ | $2.108 \times 10^{-9}$ | $8.724 \times 10^{-6}$ | $3.284 \times 10^{-6}$ | $1.132 \times 10^{-12}$ |

We present the following results of $\lambda = 0.4, 0.5$ using weak and exact asymptotics: see Figure 1 for the equation residual of the solution evaluated at the final training iterate and Figure 2 for the evolution of the losses. We are able to achieve high accuracy over a large domain, but using exact asymptotics is preferred for both the smooth and nonsmooth case of $\lambda$.

## 4.2 Boussinesq Equation

For the 2D Boussinesq equation on the half plane $y_2 \geq 0$, in vorticity form with the self-similar ansatz, we get the following profile equations for $(\Omega, U_1, U_2, \Phi, \Psi)$ as in Wang et al. (2023):

$$\Omega + ((1+\lambda)(y_1, y_2)^T + (U_1, U_2)^T) \cdot \nabla\Omega = \Phi,$$
$$(2 + \partial_{y_1}U_1)\Phi + ((1+\lambda)(y_1, y_2)^T + (U_1, U_2)^T) \cdot \nabla\Phi = -\partial_{y_1}U_2\Psi,$$
$$(2 + \partial_{y_2}U_2)\Psi + ((1+\lambda)(y_1, y_2)^T + (U_1, U_2)^T) \cdot \nabla\Psi = -\partial_{y_2}U_1\Phi,$$
$$\partial_{y_1}U_1 + \partial_{y_2}U_2 = 0, \quad \Omega = \partial_{y_1}U_2 - \partial_{y_2}U_1, \quad \partial_{y_1}\Psi = \partial_{y_2}\Phi,$$

where $(\Omega, U_1, \Phi)$ are odd and $(U_2, \Psi)$ are even in $y_1$.

For the boundary conditions, we impose a non-penetration boundary condition $U_2(y_1, 0) = 0$ along with decaying weak asymptotics at the far field, with Dirichlet boundary conditions $\Phi = \Psi = 0$ and Neumann boundary conditions for the velocity field $\nabla(U_1, U_2)^T = 0$. For the nondegeneracy condition, we impose $\partial_{y_1}\Omega(0, 0) = -1$ and use Taylor expansion to enforce a hard constraint as in Subsection 3.2. We find that enforcing a hard constraint is much more effective to avoid converging to a trivial solution than enforcing soft constraints.

For each function, we use a 7-layer MLP with width 30, hard constraints on parity, and activation function $\mathrm{SiLU} = \frac{x}{1+e^{-x}}$ to better model the growth at the far field. For sampling, we sample 1000 points on each boundary of the square $(z_1, z_2) \in [0, 30]^2$, and 5000 points each for the interior and smoothness losses, where we sample $(z_1, z_2)$ with equal probability uniformly on $[0, 30]^2$ and $[0, 5]^2$ for the interior loss and with equal

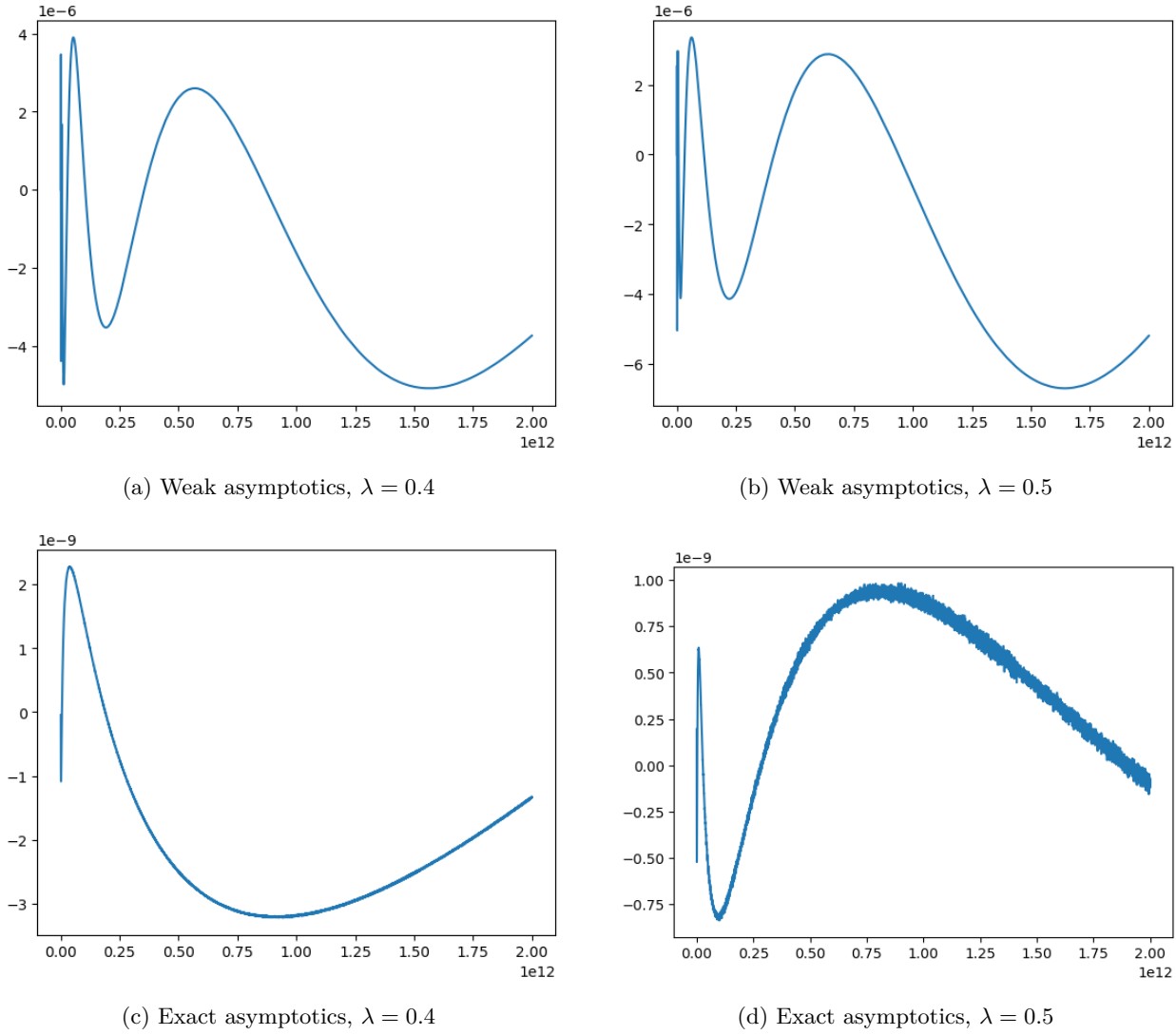

(a) Weak asymptotics, $\lambda = 0.4$

(b) Weak asymptotics, $\lambda = 0.5$

(c) Exact asymptotics, $\lambda = 0.4$

(d) Exact asymptotics, $\lambda = 0.5$

Figure 1: **Pointwise residual for the 1D Burgers profile equation on a large unbounded-domain test grid.** We plot the final residual $R(y)$ after training on $z \in [0, 30]$, corresponding to $y \in [0, 5 \times 10^{12}]$. The top row uses weak Neumann far-field asymptotics, while the bottom row uses the exact far-field asymptotics $g(y)$ with the cutoff parametrization described in the text. Since the architecture, optimizer, hard parity constraint, sampling strategy, and training budget are fixed across panels, this figure isolates the effect of the far-field parametrization.

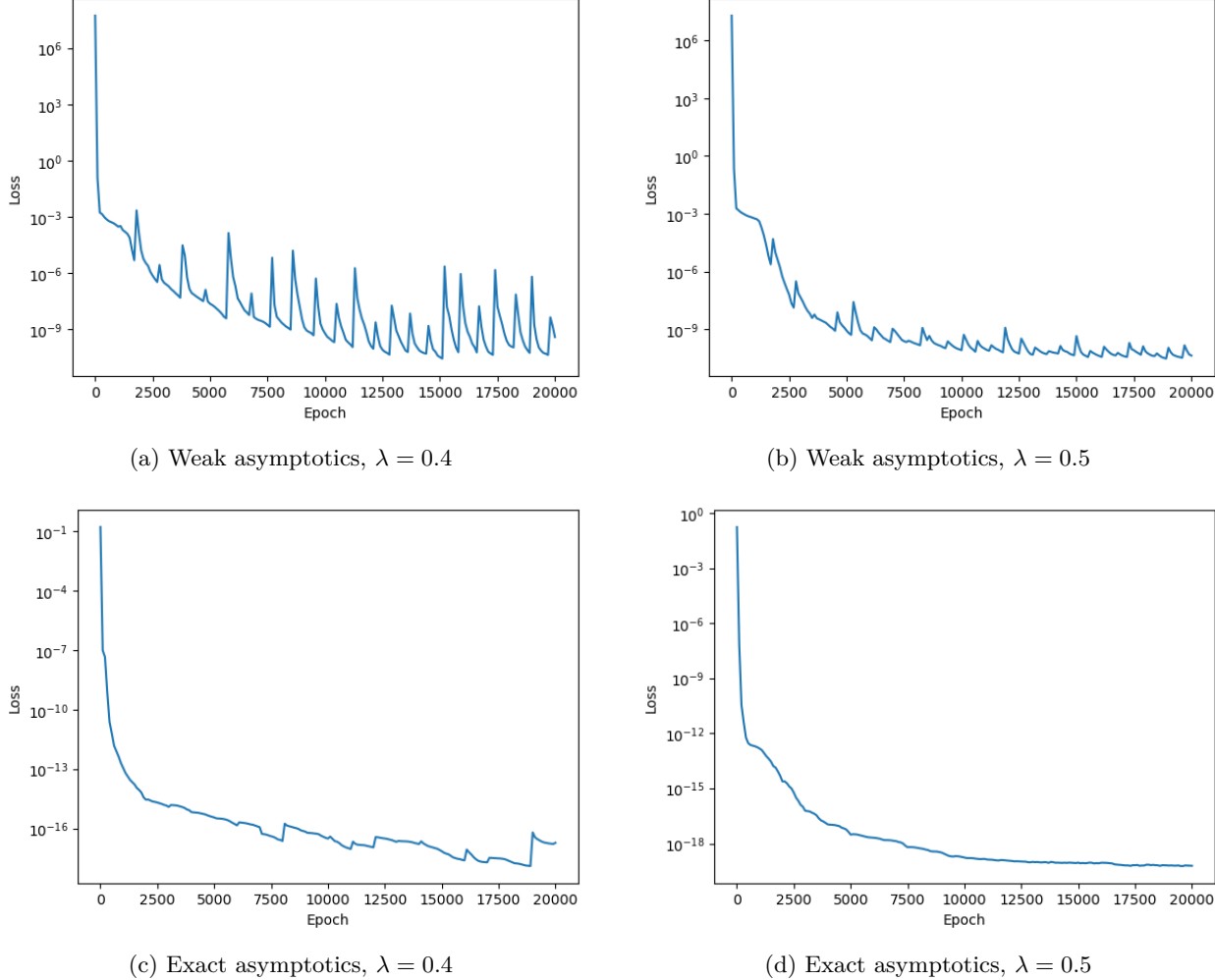

(a) Weak asymptotics, $\lambda = 0.4$

(b) Weak asymptotics, $\lambda = 0.5$

(c) Exact asymptotics, $\lambda = 0.4$

(d) Exact asymptotics, $\lambda = 0.5$

Figure 2: **Training-loss trajectories for the 1D Burgers far-field ablation.** We plot the total loss $\mathcal{J}$ from Eq. (2) over 20000 SSBroyden1 iterations. The top row uses weak asymptotics and the bottom row uses exact asymptotics, with all other components fixed. The visible jumps are caused by the periodic resampling of collocation points every 1000 iterations. Exact asymptotics gives faster convergence and a substantially lower final loss for both the nonsmooth case $\lambda = 0.4$ and the smooth case $\lambda = 0.5$.

probability uniformly on $[0,3]^2$ and $[0,0.5]^2$ for the smoothness loss, ensuring smoothness near the origin. Again, we are computing effectively in a large domain $[0, 5 \times 10^{12}]^2$ in the $y$ variables.

For optimization, we use Adam for 10000 epochs with resampling, followed by the optimizer SSBroyden 1 as in Urbán et al. (2025) with 40000 epochs and resampling every 1000 epochs. The learning rate of Adam is set to be 0.001 for the functions and 0.1 with $\beta = (0.9, 0.9)$ for $\lambda$, with a decay of 0.9 after 5000 epochs.

We present the final profiles and the residual of the PDEs near the origin respectively in Figure 3 and Figure 4, and the evolution of losses in Figure 5. We can see that the residual is small throughout the domain plotted, especially well-addressed at the origin. Compared to Wang et al. (2023), our reported training loss is about four digits smaller and the reported equation residual is about two digits smaller. We do not present this as a fully controlled side-by-side benchmark, because the original implementation, collocation points, and exact evaluation protocol of Wang et al. (2023) are not publicly available. Instead, we use it as a quantitative reference point. In our Boussinesq runs, Adam and L-BFGS are relatively cheap per step but plateau at a residual level comparable to the earlier PINN computation. The self-scaled Broyden stage is more expensive, but it is the only stage in our experiments that continues reducing the residual beyond this plateau. The full Boussinesq run uses about 10 days of CPU time on a MacPro 2019 with a 2.5GHz 28-core Intel Xeon W processor.

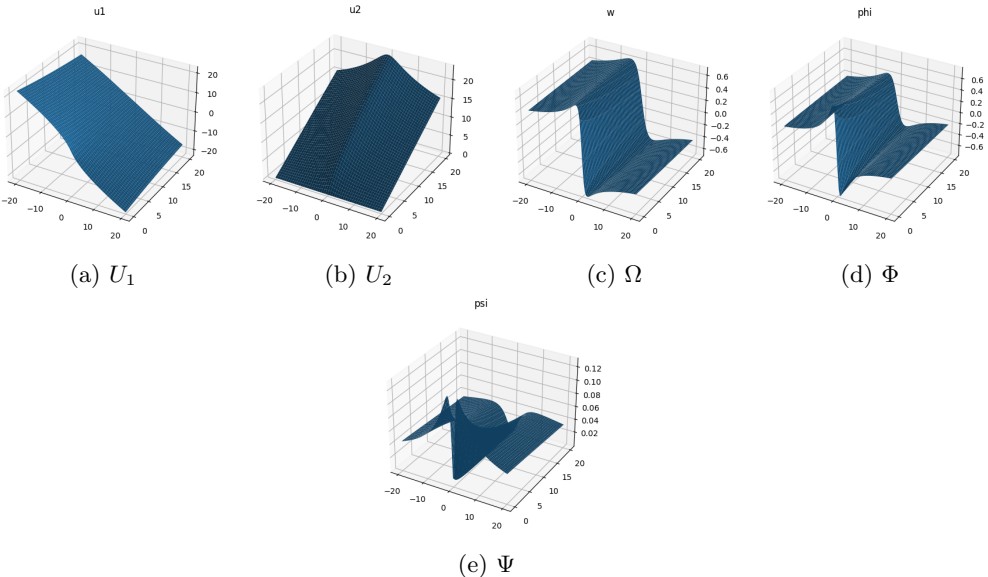

(a) $U_1$      (b) $U_2$      (c) $\Omega$      (d) $\Phi$

(e) $\Psi$

Figure 3: **Final learned profiles for the 2D Boussinesq self-similar system.** The panels show the five neural-network-represented fields $(U_1, U_2, \Omega, \Phi, \Psi)$ on the plotted $(z_1, z_2)$ grid. The parity constraints are built into the parametrization: $(\Omega, U_1, \Phi)$ are odd in $y_1$, while $(U_2, \Psi)$ are even in $y_1$.

## 5 Conclusions and Future Work

We demonstrate the importance of enforcing appropriate asymptotics, enforcing hard constraints, and adopting a better optimizer for solving PDEs using neural networks on an infinite domain. We achieve better accuracy for problems crucial to the study of singularity formulations. As a future direction, we believe a better enforcement of far-field asymptotics, formulated as hybrid asymptotics, might have the potential of driving PDE residual to machine precision, potentially amenable to rigorous computer-assisted proofs using the profiles identified by the neural networks. Another direction is to use PINO, the idea of operator learning on a range of scaling parameters, to learn a collection of profiles with different scalings.

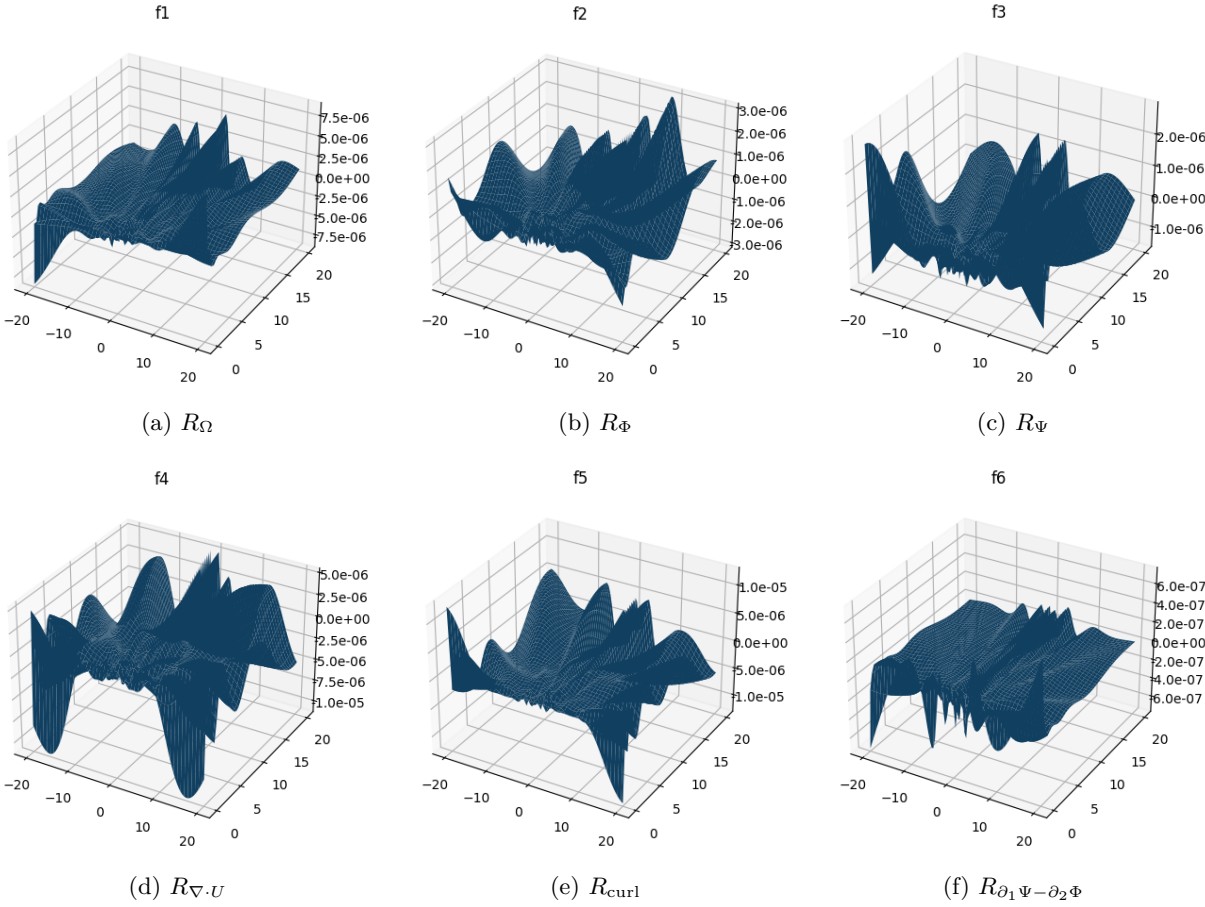

Figure 4: **Pointwise residuals for the six equations in the 2D Boussinesq profile system.** The panels correspond, in order, to the residuals of the vorticity equation, the $\Phi$ equation, the $\Psi$ equation, incompressibility, vorticity–velocity consistency, and the compatibility condition $\partial_{y_1}\Psi = \partial_{y_2}\Phi$. All residuals are evaluated after the final SSBroyden1 iterate on the plotted $(z_1, z_2)$ grid near the origin, where the smoothness loss is sampled most densely and the hard Taylor constraint enforces $\partial_{y_1}\Omega(0,0) = -1$.

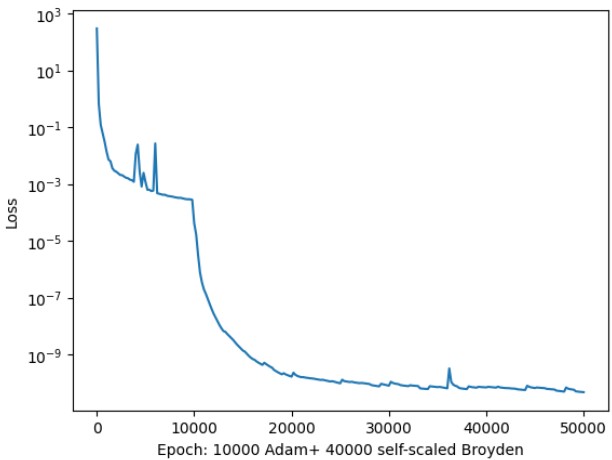

Figure 5: **Training-loss trajectory for the 2D Boussinesq profile.** We plot the total loss $\mathcal{J}$ from Eq. (2) over the full optimization run. The first 10000 iterations use Adam with resampling, after which the optimizer is switched to SSBroyden1 for 40000 iterations with resampling every 1000 iterations. The sharp decrease after the switch illustrates the role of the self-scaled Broyden stage in reaching the final high-precision residual.

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
