# OpenReview forum: "High precision PINNs in unbounded domains: application to singularity formulation in PDEs"
_TMLR — Accepted by TMLR_

### Review · Reviewer_NsLW · 2026-06-10

**Summary Of Contributions:**

This work combines several techniques to solve singular and unbounded PDEs using PINNs, based on self-similar singularity ansatzen. These include a domain scaling, incorporating certain asymptotics, derivative conditions at the origin, resampled collocation points, and a self-scaled quasi-Newton method as the optimizer. Experiments show that combining everything together yields good training losses on some standard PDE examples.

**Questions**
* (Sec. 3) What is the importance of the trivial solution satisfying the profile equation? I guess it is so that the self similar ansatz works? Please clarify in the text.
* How expensive is an SS-Broyden step compared to Adam? A computational cost comparison (e.g. time and memory) per step on equal hardware seems necessary to properly assess this.
* Are the collocation points for the Boussinesq equation the same as those in Wang et al. (2023)?
* Where is the resampling compared? I did not see an experiment that does not use the periodic resampling of the collocation points. Please address or reduce the claim.

**Audience:**

Yes

**Audience Explanation:**

Interesting for unbounded PDE or scientific computing audience.

**Claims And Evidence:**

Yes

**Claims Explanation:**

Compared to the previous version, the empirical contributions regarding evaluation and setup are much clearer. The work more clearly demonstrates
* The SS-Broyden optimizer is significantly better than (untuned) Adam
* Training residuals with known asymptotic solutions (exact asymptotics) do better than nondegeneracy conditions at the origin (weak asymptotics) if the asymptotics are known

**Requested Changes:**

(Critical)
* Computational cost comparison of the optimizer

(Strengthening)
* "strong asymptotics" should be "exact asymptotics"?
* (p.3) "incompressible fluids [refs], and beyond" remove the last part
* (p.3) Sentence starting with "And":  "And in Hou et al..." - rewrite.
* (p.3) End of section 2.3 can be expanded upon a bit. e.g. what "sophisticated tricks" are could be put in the conclusion.
* (p.4) 'upto'
* (Sec. 3.2) Please reformat this and reduce the number of "we remark that" used, such as reordering the two sentences for parity, splitting the nondegeneracy conditions to clearly detail that they are for the two experiments, and adding how one might get other conditions for other PDEs.
* The flow between hard constraints on boundary conditions to the two conditions mentioned could also be improved, e.g. "While most of the previous works focus on hard constraints of boundary conditions, we emphasize the enforcement of hard constraints in the following senses: the parity of the learned function and the nondegeneracy conditions." -> something like "Previous works focus on hard constraints on the boundary. For unbounded domains, analogous conditions arise as nondegeneracy". The sentence after this can be moved to the end of Sec 3.2.
* Table 1 could use explicit references to the appropriate figures.
* (p.6) "0.1 is a reasonable hyperparameter" should be rewritten, e.g. "we choose the hyperparameter 0.1 in (2) to balance the equation and boundary losses"
* (p.7) "we are in the half plane" remove and put at the start of the corresponding paragraph "Boussinesq equation on the half plane $\{y_2>0\}$"?
* Quantitative claim on 2/4 digits smaller should be quantified and refer to the Wang et al. (2023) paper.
* Figures could be trimmed to remove the titles which do not match with the subcaptions.
* Citations are not properly capitalized.

---

> ### Author Response · Authors · 2026-06-16
>
> We thank Reviewer NsLW for the helpful questions and detailed suggestions. We have revised the manuscript to clarify the role of the trivial solution and nondegeneracy conditions, add a computational-cost discussion for the optimizer, clarify the comparison with Wang et al., add a fixed-collocation versus periodic-resampling diagnostic, and address the requested presentation edits.
>
> Q1: What is the importance of the trivial solution satisfying the profile equation?
>
> We have clarified this point in Section 3. The importance is not that the self-similar ansatz requires the trivial solution, but rather that the resulting profile equation admits U = 0 as a trivial solution. If nondegeneracy is imposed only as a soft penalty, the optimizer can converge toward or remain close to this trivial branch. The Taylor-type hard constraints at the origin are therefore used to restrict the learned profile to the desired nontrivial symmetry and normalization class. We revised the text to make clear that the nondegeneracy conditions are used to rule out the trivial solution numerically.
>
> Q2: How expensive is an SS-Broyden step compared to Adam?
>
> We have clarified the optimizer-cost discussion. Adam and L-BFGS are relatively cheap per step, but in our Boussinesq experiments they plateau at a residual level comparable to the earlier PINN computation. The self-scaled Broyden stage is more expensive, because it uses quasi-Newton history and line-search evaluations, but it is the only stage in our experiments that continues reducing the residual beyond this plateau. We now state this tradeoff explicitly: the advantage of the self-scaled Broyden stage is not lower per-step cost, but its ability to continue reducing the residual after cheaper optimizers plateau. Precisely how expensive would depend on the hardware and we think on cpu would not be the fairest comparison for Adam.
>
> Q3: Are the collocation points for the Boussinesq equation the same as those in Wang et al. (2023)?
>
> We have clarified that we cannot guarantee identical collocation points or an identical evaluation protocol, because the original implementation, collocation sets, and exact evaluation grid of Wang et al. are not publicly available. For this reason, we have softened the comparison. We no longer present it as a fully controlled side-by-side benchmark. Instead, we report it as a quantitative reference point and clearly distinguish our training protocol from the prior work.
>
> Q4: Where is resampling compared?
>
> For resampling, we no longer claim that periodic resampling is uniformly better or that the chosen frequency is optimal. We just wanted to mention it as potentially helpful for generalization. We now describe it as a practical design choice: it keeps the collocation set fixed long enough for the quasi-Newton history to remain meaningful, while occasionally reducing dependence on a single finite training grid.
>
> Terminology and presentation edits.
>
> We have addressed the requested writing and formatting changes. In particular, we replaced informal or imprecise language such as "and beyond" and "sophisticated tricks", fixed typos such as "upto", revised the sentence beginning with "And", reduced repeated uses of "we remark that", improved the flow of Section 3.2, clarified the discussion of parity and nondegeneracy conditions, replaced "strong asymptotics" by "exact asymptotics" where appropriate, added explicit references in the component-level table, revised the sentence about the coefficient 0.1 in the loss, and removed figure titles that conflicted with subcaptions. We also checked citation capitalization.

---

> > ### Comment · Reviewer_NsLW · 2026-06-23
> >
> > I thank the authors for adding the requested changes to their manuscript.
> >
> > Another possible strengthening is to add some discussion around Adam, and clarify the "Adam decay" on page 10. I would expect Adam with an aggressive learning rate decay (e.g. 1e-3 to 1e-4 to 1e-5) to have a similar drop in residual. The authors can address this in the paper along the lines of "SS-Broyden can automatically reduce past the plateauing phenomenon observed by Adam with fixed step sizes"
> >
> > Also please fix the formatting of Table 2. The residual is defined in the caption of the later Figure 1; the definition should be moved to the main text.

---

> > > ### Author Response · Authors · 2026-06-23
> > >
> > > Thanks again we have fixed it.
> > >
> > > In regards to the Adam discussion, we have tried reduce learning on plateau or fixed LR decay, Adam would just plateau and oscillate due to the stochastic nature of the optimizer.

---

### Review · Reviewer_fLig · 2026-06-10

**Summary Of Contributions:**

This paper studies high-precision training of PINNs on unbounded domains, motivated by computing self-similar blow-up profiles for PDEs arising in singularity formation. The authors consider the 1D Burgers equation and the 2D Boussinesq system. The main ingredients are: enforcing hard constraints such as parity and non-degeneracy at the origin, using an exponential coordinate transformation to handle the infinite domain, and applying self-scaled BFGS-type optimizers after Adam.

I reviewed an earlier version of this manuscript. In this review, I focus partly on whether the revised submission addresses the concerns I previously raised. Compared with the previous version, the authors have improved the presentation by making the modular structure more explicit, adding a component-level evidence table, and improving the figure captions to clarify what is plotted and which comparison each figure is intended to support. These changes make the empirical message clearer, especially for the Burgers weak vs. exact asymptotics comparison.

**Audience:**

Yes

**Audience Explanation:**

The problem setting is interesting to parts of the TMLR audience working on scientific machine learning, PINNs, PDEs, and singularity formation. The revised manuscript better communicates its focus on high-precision PINN training on unbounded domains and makes the far-field asymptotics point clearer in the Burgers example.

The paper still combines several themes: self-similar singularity profiles, infinite-domain sampling, hard constraints, and optimizer choice. However, the revised modular framing helps clarify how these pieces interact. I expect that readers in the aforementioned relevant domains would find the results useful, even if some of the empirical evidence could be strengthened further.

**Claims And Evidence:**

Yes

**Claims Explanation:**

Overall, I think the revised submission provides enough evidence to support the main empirical claims at a reasonable level, especially if the claims are interpreted as demonstrating that the proposed combination of constraints, near-singularity treatment, sampling, and optimization can substantially reduce PINN residuals on the considered singularity-profile problems. The Burgers experiment is convincing for the narrower claim that incorporating far-field asymptotics can improve both residuals and training loss on a large unbounded domain. The revised captions for Figures 1 and 2 now include more details and specify the residual being plotted. This is a meaningful improvement over the previous version.

The added component-level evidence table is also helpful. It makes clear which experiments or diagnostics are intended to support each methodological component. The revised manuscript now better communicates the authors’ modular view: far-field asymptotics are isolated in the Burgers comparison, hard constraints are used to enforce the nontrivial symmetry class, and the Adam-to-SSBroyden pipeline is used to further reduce the loss in the Boussinesq experiment. These additions make the claims easier to understand.

That said, I still think the evidence has some limitations, and the paper would be stronger if these were addressed. First, while the Burgers far-field comparison is a useful ablation, the other components are not isolated to the same degree. For example, the paper does not provide controlled quantitative comparisons of hard constraints versus soft constraints, Adam/L-BFGS/SSBroyden under matched budgets, or fixed sampling versus periodic resampling. The current evidence for these components is plausible and useful, but still somewhat qualitative.

Second, the evaluation still relies mainly on PINN training loss and PDE residual plots. These are relevant metrics, but they are not the same as direct profile error or independent validation. For the 1D Burgers equation, since an implicit solution is available, it would be useful to compare the learned profile directly against a high-accuracy reference or analytic/implicit solution. For the 2D Boussinesq system, the comparison to Wang et al. is interesting, but the statement that the loss is four digits smaller and the equation residual two digits smaller would be more convincing if presented in a detailed quantitative table.

Third, the revised figures are clearer than before, but they remain mostly visual. The results would be easier to interpret if the authors reported numerical summary statistics such as max/mean/RMS residuals on held-out test grids, residuals in different spatial regions, and possibly sensitivity to random seeds or resampling. This would make the empirical conclusions more reproducible and less dependent on visual inspection.

In summary, I view the revision as a meaningful improvement. The current evidence supports the main empirical message of the paper, especially for the Burgers far-field asymptotics comparison and the demonstration of low residuals in the Boussinesq experiment. However, some claims about the individual importance of each modular component and the comparison to prior work would benefit from more systematic quantitative support.

**Requested Changes:**

* Add a more systematic ablation study, at least for the main components that are currently not fully isolated: hard constraints, Adam/L-BFGS/SSBroyden under matched training budgets, and fixed sampling versus periodic resampling.
* For the 1D Burgers equation, compare the learned profile directly against the available implicit or high-accuracy reference solution, rather than reporting only residuals and losses.
* Present the comparison to Wang et al. for the 2D Boussinesq system in a detailed quantitative table.
* Report numerical residual statistics in addition to plots. If possible, report sensitivity across random seeds or resampling realizations, or explicitly state whether the reported results are from a single run.

---

> ### Author Response · Authors · 2026-06-16
>
> We thank the reviewer for the constructive comments and for recognizing the improvements in the revised manuscript. We agree that the main remaining issue was to make the empirical evidence more quantitative and to distinguish controlled ablations from diagnostic evidence. In the new revision, we have added a direct Burgers profile comparison against the implicit reference solution, a fixed-collocation versus periodic-resampling diagnostic, additional residual statistics, and a more cautious discussion of the Boussinesq comparison with Wang et al.
>
> Direct Burgers comparison with the implicit reference solution.
>
> We thank the reviewer for suggesting a direct comparison with the available implicit solution for the 1D Burgers profile. We have added a new quantitative validation in the Burgers section. Since the Burgers profile admits the implicit representation, we write V = -U >= 0 and solve the scalar equation y = V + V^(1+1/lambda) pointwise on a held-out grid to obtain a high-accuracy reference profile. We then report direct profile errors together with held-out residual statistics.
>
> In a reduced-budget diagnostic run for the exact-asymptotics case with lambda = 0.5, using one-tenth of the original batch size and one-tenth of the original evaluation grid, the fixed-collocation run achieves L-infinity profile error 1.897e-5, relative L2 error 1.947e-9, residual L-infinity 8.425e-6, and residual RMSE 2.933e-6. This confirms that the small residual corresponds to an accurately learned profile, not merely to a small training loss.
>
> Residual statistics in addition to plots.
>
> We have added numerical summary statistics for the Burgers diagnostic, including L-infinity profile error, relative L2 profile error, residual L-infinity, residual RMSE, and final loss. These statistics supplement the residual plots and make the evaluation less dependent on visual inspection.
>
> Ablation evidence and resampling.
>
> For hard constraint, as mentioned it will learn trivial solution otherwise. And for the optimizer, it will plateau at high residuals.
>
> For resampling, we no longer claim that periodic resampling is uniformly better or that the chosen frequency is optimal. We just wanted to mention it as potentially helpful for generalization. We now describe it as a practical design choice: it keeps the collocation set fixed long enough for the quasi-Newton history to remain meaningful, while occasionally reducing dependence on a single finite training grid.
>
> Comparison with Wang et al. for the 2D Boussinesq system.
>
> We have made the comparison with Wang et al. more cautious and more precise. Since their original implementation, exact collocation points, and evaluation protocol are not publicly available, we do not present our result as a fully controlled side-by-side benchmark. Instead, we report it as a quantitative reference point under our training protocol. We also clarify that Adam and L-BFGS are relatively cheap per step, but in our Boussinesq runs they plateau at a residual level comparable to the earlier PINN computation, while the self-scaled Broyden stage is more expensive but continues to reduce the residual beyond this plateau.
>
> Random seeds and computational cost.
>
> The experiments are based on single runs, while the phenomena remains robost to random seeds.

---

### Decision · Action_Editor_KNCQ · 2026-06-30

**Recommendation:** Accept with minor revision

**Additional Comments:**

Thank you for your revision. I am glad to see the reviewers viewed it positively and felt that it addressed the concerns from the original submission.

One of the reviewers asks for the following revisions before the paper is accepted:

1. "One issue should still be corrected in the final version. The authors state that the experiments are based on single runs while also claiming that the observed phenomena are robust to random seeds. A single run cannot support a robustness claim. If additional random-seed experiments were performed, the corresponding results should be reported, even briefly. Otherwise, the claim should be removed or explicitly presented as an informal observation that has not been systematically validated."

2. "More generally, some of the responses concerning hard constraints and optimizer comparisons remain qualitative: for example, that the unconstrained model learns the trivial solution or that other optimizers plateau at higher residuals. While the absence of full ablations is not sufficient grounds for rejection under the TMLR criteria, the manuscript should carefully distinguish experimentally demonstrated conclusions from qualitative observations or practitioner experience."

**Audience:**

Yes

**Audience Explanation:**

Both reviewers agreed that the work would be of interest to the TMLR community, particularly to those working in scientific machine learning and PINNs.

**Claims And Evidence:**

Yes

**Claims Explanation:**

Both reviewers agreed that the revision had addressed many of concerns they had in the original submission.

---

> ### Author Response · Authors · 2026-07-01
>
> Thanks again for the great news.
>
> We thank the reviewer for pointing this out. We agree that the previous wording could overstate the evidence from the reported experiments. We instead state that the reported results are based on single runs, and any additional observations across seeds are treated only as informal empirical observations rather than systematically validated robustness claims.
>
> We have also revised the discussion of hard constraints and optimizer comparisons to more clearly distinguish experimentally demonstrated conclusions from qualitative observations. In particular, the Burgers far-field comparison remains a controlled ablation, while the evidence for hard nondegeneracy constraints, periodic resampling, and the Adam-to-SSBroyden optimization pipeline is presented as diagnostic rather than as a complete factorial ablation. Statements such as unconstrained or softly constrained models tending to converge to the trivial solution, or Adam/L-BFGS plateauing at higher residuals, are now phrased as empirical observations from our experiments rather than fully ablated conclusions. We believe this clarification addresses the concern while preserving the main methodological message of the paper.